# Innovative Biosensing Approaches for Swift Identification of *Candida* Species, Intrusive Pathogenic Organisms

**DOI:** 10.3390/life13102099

**Published:** 2023-10-22

**Authors:** Dionisio Lorenzo Lorenzo-Villegas, Namra Vinay Gohil, Paula Lamo, Swathi Gurajala, Iulia Cristina Bagiu, Dan Dumitru Vulcanescu, Florin George Horhat, Virgiliu Bogdan Sorop, Mircea Diaconu, Madalina Ioana Sorop, Andrada Oprisoni, Razvan Mihai Horhat, Monica Susan, ArunSundar MohanaSundaram

**Affiliations:** 1Faculty of Health Sciences, University Fernando Pessoa-Canarias, 35450 Santa Maria de Guia, Spain; dlorenzo@ufpcanarias.es; 2Department of Internal Medicne, Medical College Baroda, Vadodara 390001, India; namragohil1996@gmail.com; 3Department of Internal Medicne, SSG Hospital Vadodara, Gotri, Vadodara 390021, India; 4Escuela Superior de Ingeniería y Tecnología, Universidad Internacional de La Rioja, 26006 Logroño, Spain; paula.lamo@unir.net; 5College of Applied Medical Sciences in Jubail, Imam Abdulrahman bin Faisal University, Dammam 31441, Saudi Arabia; dr.gurajalaswathi@gmail.com; 6Department of Microbiology, “Victor Babes” University of Medicine and Pharmacy, Eftimie Murgu Square 2, 300041 Timisoara, Romania; dan.vulcanescu@umft.ro (D.D.V.); horhat.florin@umft.ro (F.G.H.); 7Multidisciplinary Research Center on Antimicrobial Resistance (MULTI-REZ), Microbiology Department, “Victor Babes” University of Medicine and Pharmacy, Eftimie Murgu Square 2, 300041 Timisoara, Romania; 8Department of Obstetrics and Gynecology, “Victor Babes” University of Medicine and Pharmacy, Eftimie Murgu Square, No. 2, 300041 Timisoara, Romania; bogdan.sorop@gmail.com (V.B.S.); diaconu.mircea@umft.ro (M.D.); 9Doctoral School, “Victor Babes” University of Medicine and Pharmacy, 300041 Timisoara, Romania; pop_madalina_91@yahoo.ro; 10Department of Pediatrics, Discipline of Pediatric Oncology and Hematology, “Victor Babes” University of Medicine and Pharmacy, Eftimie Murgu Square, No. 2, 300041 Timisoara, Romania; oprisoni.licinia@umft.ro; 11Department of Conservative Dentistry and Endodontics, Faculty of Dental Medicine, “Victor Babes” University of Medicine and Pharmacy Timisoara, Eftimie Murgu Square 2, 300041 Timisoara, Romania; razvanhorhat@yahoo.com; 12Centre for Preventive Medicine, Department of Internal Medicine, “Victor Babes” University of Medicine and Pharmacy, Eftimie Murgu Square, No. 2, 300041 Timisoara, Romania; monisusan@yahoo.com; 13School of Pharmacy, Sathyabama Institute of Science and Technology, Jeppiaar Nagar, Rajiv Gandhi Salai, Chennai 600119, India; arunsundar.pharmacy@sathyabama.ac.in

**Keywords:** *Candida*, biosensor, diagnostic, fungi, rapid detection

## Abstract

*Candida* is the largest genus of medically significant fungi. Although most of its members are commensals, residing harmlessly in human bodies, some are opportunistic and dangerously invasive. These have the ability to cause severe nosocomial candidiasis and candidemia that affect the viscera and bloodstream. A prompt diagnosis will lead to a successful treatment modality. The smart solution of biosensing technologies for rapid and precise detection of *Candida* species has made remarkable progress. The development of point-of-care (POC) biosensor devices involves sensor precision down to pico-/femtogram level, cost-effectiveness, portability, rapidity, and user-friendliness. However, futuristic diagnostics will depend on exploiting technologies such as multiplexing for high-throughput screening, CRISPR, artificial intelligence (AI), neural networks, the Internet of Things (IoT), and cloud computing of medical databases. This review gives an insight into different biosensor technologies designed for the detection of medically significant *Candida* species, especially *Candida albicans* and *C. auris*, and their applications in the medical setting.

## 1. Introduction

Fungal pathogens had long been relegated to the “dungeons” of the modern healthcare system as compared to bacterial pathogens. However, the outbreaks of infections by *Candida* species alongside SARS-CoV-2 have brought the problem to the foreground. Even the World Health Organization (WHO) was compelled in late 2022 to publish the fungal priority pathogens list (FPPL) for the first time—a catalog of 19 fungi that threaten public health [1]. The Global Action for Fungal Infections (GAFFI) factsheet estimates that more than 300 million people around the globe are afflicted with one serious fungal infection whereas 25 million are at high risk of dying [2].

The medically important species of the *Candida* spectrum include *Candida albicans*, *Candida auris*, *Candida krusei*, *Candida tropicalis*, *Candida parapsilosis*, *Candida rugosa*, *Candida guilliermondii*, and *Candida glabrata* [3]. *C. albicans* is a commensal microbe and part of normal human flora, residing harmlessly on the skin and inside the body, in locations such as the mouth, gut, throat, and vagina. However, it becomes an opportunistic invader in immunocompromised or immunodeficient individuals. An invasive disease caused by any *Candida* species is termed candidiasis. Candidiasis may occur in individuals due to waning immunity, pH change, diabetes, surgical procedures, antibiotic therapy, neutropenia, parenteral nutrition, malignancy, or HIV, and in neonates [4,5]. Oropharyngeal candidiasis or thrush occurring in the mouth and throat, characterized by thick, white patches in the oral cavity, is common in people wearing dentures, with HIV history, with diabetes, and under strong medications; esophageal candidiasis is common in AIDS patients. The most common genitourinary candidiasis is vaginal candidiasis or vulvovaginal candidiasis. This occurs very commonly in females due to medicines or hormonal or immune system imbalances. The symptoms are characterized by vulvovaginal inflammation like redness, soreness, irritation, itching, burning, and thick cheesy discharge. As per the estimations, about 75% of all adult healthy women experience at least one bout of infection in their lifetime, with at least half of them becoming reinfected [6].

Bloodstream *Candida* infections are quite common in hospitalized patients. The mortality rate among candidemia patients ranges from ~20 to 40% [7,8,9,10]. *C. auris*, a multidrug-resistant pathogen, causes serious bloodstream and wound infections in hospitalized patients, especially those having breathing tubes, feeding tubes, or central venous catheters [11].

Azoles, echinocandins, and amphotericin B drugs are commonly used to treat *Candida* infections. However, resistance patterns are on the rise, with *C. auris* being among the most significant multidrug-resistant species, while *C. krusei* and *C. glabrata* are intrinsically resistant to echinocandins [4,12,13,14,15]. Therefore, early detection is crucial to mitigate drug resistance emergence.

The timeliness of therapy is a victim of insensitive and slow diagnostic tools like the fungal culture method, microscopy, serology, and histopathology [12]. This delay leads to the emergence of antifungal resistance, incurring huge medical costs. Thus, rapid and specific pathogen identification is the key to a prompt and effective treatment strategy. The evolution of culture-independent, molecular, and proteomic technology-based modern diagnostics can assist in species identification and detection. These are helpful in multiplex detection and provide information on the drug resistance profile of each fungal pathogen. Some of these technologies hold the key to be used as point-of-care devices. These approaches target specific biomarkers of the pathogen.

Sudden fungal outbreaks can be tackled with revolutionary concepts in the field of biosensors with sensitivity down to picomolar (10^−12^ M), femtomolar (10^−15^ M), and attomolar ranges (10^−18^ M). Hence, next-generation approaches based on molecular techniques such as polymerase chain reaction (PCR), DNA-sequencing-based methods, protein fingerprinting by matrix-assisted laser desorption/ionization–time-of-flight mass spectrometry (MALDI-TOF-MS), and ultrasensitive laser-based technologies that target specific Candidal biomarkers [12] are commonly used techniques nowadays. Alongside wearable sensors, artificial intelligence, cloud computing, data management, the Internet of Things (IoT), biomedical database management, highly skilled personnel, and decision-makers^.^ will aid in this direction [16]. The fundamentals of computer science and technology—artificial intelligence and neural networks—have crept into the field of biosensors too. A biosensor is a bio-analytical device used for detection purposes. Micro-electro-mechanical system (MEMS) cantilever biosensors are a class of prospective biosensors with high levels of miniaturization, embedded microelectronics, intelligence, mass production, and low budget [17]. A micromechanical cantilever array sensor was designed to monitor yeast hyphal growth and selective immobilization in real time using microbeam biosensors as early as 2005 [18].

Traditional diagnostic methods for *Candida* infections, such as blood culture and microscopy, have several limitations that can lead to delayed or inaccurate diagnosis. These methods are time-consuming, require skilled personnel, and may not be sensitive enough to detect low levels of *Candida* species cells in clinical samples [19]. In addition, species identification using traditional methods can be challenging due to the high degree of phenotypic similarity between different *Candida* species [20]. Furthermore, conventional methods may not be able to detect antifungal resistance, which is becoming an increasingly important issue in the management of *Candida* infections [21]. Therefore, there is a need for more innovative, rapid, and accurate approaches for the detection and identification of *Candida* species. Biosensors are the appropriate tools in this approach. Generally speaking, biosensors offer rapid and accurate detection of *Candida* species, which can lead to earlier diagnosis and treatment; can be more cost-effective than traditional diagnostic methods; can be used for real-time monitoring of circulating analytes, which can reflect the efficacy of antifungal treatment; and can be designed to be portable and easy to use, making them suitable for point-of-care diagnosis [22,23].

This review aims to give an update and call for more attention in regard to medically important *Candida* surveillance by different types of biosensors, focusing on the available literature published until now to the best of our knowledge, emphasizing the impact of modern technologies and their applications in the medical setting.

## 2. *Candida* Biomarkers

Diagnostic and commercial tests for invasive candidiasis have been developed, based on serological markers like mannan antigen, anti-mannan antibodies [24], the anti-mycelium/germ tube antibody (CAGTA) test [25], and (1 → 3), β-d-glucan (BDG). Platelia™ and Platelia™ *Candida* antibodies are applied for detecting mannan antigen and anti-mannan antibodies, respectively, in infected serum [24]. Potential *Candida* biomarkers exploited for diagnostic tests are depicted in Figure 1.

Mannan is a major cell wall component and is the primary circulating antigen encountered during infection. Enolase [26] and arabinitol [27], antibodies against chitin (ACCA) and synthetic disaccharide fragments of glucans (ALCA) [28], have also been reported as marker molecules.

## 3. Conventional Diagnostic Approaches

Traditional approaches for detection are mostly based on culture methods, microscopy, serology, and histology. In vitro culture methods on selective or indicator growth media such as CHROMagar™ *Candida* Plus agar prominently distinguish *Candida auris.* Direct microscopy to locate *Candida albicans* in clinical samples like sputum, swabs, tissue biopsies bronchoalveolar lavage, cerebrospinal, and other body fluids is faster as compared to the culture method. KOH, Calcoflour White, Chicago Sky blue 6B, Blankophor, Gram stain, and India ink are used for direct microscopy [24]. However, fluorescence staining shows better sensitivity [25].

The yeast, pseudohyphal, and hyphal polymorphic features of *Candida* can be identified by histology, which is a better and more cost-effective tool of choice for invasive fungal detection than the culture method. Periodic acid–Schiff (PAS) and Gomori’s methenamine silver (GMS) stains are specially used [12,26]. Sometimes the buds do not separate from each other, giving a chain-like pseudohyphal morphology [27]. Also, misidentification may occur due to similar morphology, e.g., *Histoplasma* and small *Candida* species (*C. glabrata*) [28]. Histological identification may be limited by a lack of tissue samples due to invasive processes, which may not be possible in severely sick or thrombocytopenic patients; the host’s immune status; distorted fungal morphology depending on the type of biopsy; and sample quality [12,26]. The specificity of histology can be enhanced by immunohistochemistry which may be again limited by the availability of species-specific antibodies.

Serological tests detect the presence of *Candida* antigens or circulating antibodies in the blood, like cell wall mannan glycoprotein antigen and anti-mannan antibodies. PLATELIA™ *Candida* Ag Plus 62784 (Bio-Rad Laboratories) and SERION ELISA antigen, Hemkit *Candida* Ab test detect circulating mannan antigen qualitatively and quantitatively in human serum or plasma [27]. The Wako BDG assay and Fungitell BDG assay detect the cell wall constituent, β-1,3-D-glucan, in the blood of invasive candidiasis patients [27,28]. Interference may arise in the results owing to clearance from the patient’s sera or a dip in serum antigen concentration due to antigen–antibody complexation [28].

No single technique is sufficient to provide accurate and specific information for species identification and antifungal resistance. It is better to rely on a combination of traditional and molecular techniques like PCR, sequence-based identification using peptide nucleic acid fluorescence in situ hybridization (PNA-FISH), and matrix-assisted laser desorption/ionization (MALDI) spectrometry simultaneously [12,29].

The clinical symptoms of candidiasis are non-specific and appear later in the course of infection. In addition, nearly one-third of candidiasis patients do not receive positive blood culture reports, and results are obtained at approximately 48–72 h or more. Species identification takes even longer. Its sensitivity in hematogenous disseminated candidiasis is <50% [30].

Prompt surveillance systems are of utmost importance to counter the rapid, sudden rise and spread of fungal infections worldwide, viz., *C. auris* outbreaks during the COVID-19 pandemic. Nosocomial infections, invasiveness, and high fatality rates seriously challenge the development of fast, reliable, and accurate detection systems for *Candida* species. Biosensor technology is a promising step in this direction to reduce medical costs and fatalities. Early diagnosis with biosensors will reduce medical burdens. Also, biosensors enable continuous monitoring opportunities to assess the response of the patient toward treatment modalities [12].

## 4. Biosensors—Design and Properties

A biosensor is a bio-analytical device that translates information from biological or chemical reactions into a quantifiable signal proportional to the strength of the reaction. It has three components—a biorecognition element, a transducer, and a signal processor. The quality of a biosensor depends upon its sensitivity, selectivity, reproducibility, reusability, detection limit, and response time. Sensitivity is the relationship between change in analyte concentration and signal intensity at the transducer. Selectivity is its ability to bind and respond only to the target analyte molecule and none other. Reproducibility is the ability to produce matching results in response to the target analyte each time the same/identical biosensor is used. Reusability is the ability to use the biosensor many times. The detection limit is the lowest analyte concentration that can generate a signal. Response time is the time required by the device to elicit a signal following the biochemical reaction.

Biorecognition Element: This element translates signals from a biochemical reaction involving receptor molecules and analyte molecules into a chemical or physical output signal. This is involved in the biosensing event. The input signal is proportional to the analyte concentration. Enzymes, antibodies, nucleic acids, aptamers, hormones, microbes, molecularly imprinted polymers, phage display proteins [31], and nanostructured surfaces are often exploited as biorecognition elements. Antibody- and enzyme-based biosensors offer selectivity and reusability; aptamer- and nucleic acid-based biosensors are highly sensitive to even low analyte concentrations and give reproducible results. Generally, a biosensor should be able to detect an analyte concentration within the picomolar to nanomolar range [32].

Transducer: A biosensor is categorized by the type of transducer. Transducers convert signals from the biorecognition element to measurable signals. These are connectors between the biorecognition unit and the signal processor. The biomolecule is immobilized on the transducer surface by covalent attachment, adsorption, cross-linking, entrapment, and micro-encapsulation. Transducers with high-affinity binding generally give a one-time signal, whereas low-affinity binding allows continuous monitoring of analyte concentrations [12]. Transducers may be organic, inorganic, conductors, semiconductors, or insulators [33].

Signal Processor: The signal processor converts the measurable signal from the transducer into an interpretable and quantifiable signal. The output signal from the transducer in the form of current is converted into voltage and further processed for noise removal through various filters by the processor [34].

## 5. Immobilization Techniques

The immobilization of analyte biomolecules on the sensor surface of the biorecognition element is one of the key steps of biosensor fabrication. The biomolecules may be antigens, antibodies, peptides, enzymes, nucleic acids, aptamers, or whole cells. The immobilization is essential to stabilize the biomolecules under diverse conditions of temperature, pH, hydrophobicity, oxidizing environment, and solvent polarity. This is important for the reusability of the biosensor. Immobilization brings the analyte molecules in close proximity to the transducer. The process should be such that the active site of the molecule remains accessible for interaction while maintaining bioactivity and conformational stability. Immobilization methods may be reversible or irreversible.

Reversible Immobilization: In this method, the adhered biomolecules can be detached from the surface under mild conditions to regenerate the sensor matrix for the reusability of the system. The two ways under this process are as follows:

Adsorption: It is a physical process involving hydrogen bonds, hydrophobic interaction, Van der Waal’s forces, and salt linkages. Molecules bind to the surface with low affinity, and hence the process may be reversed by variations in pH, temperature, etc., to affect the sensitivity and reproducibility. Examples include whole-cell adsorption and immunoadsorption.

Affinity Binding: Affinity binding involves non-covalent interactions between the activated support and a specific group on the biomolecule. Examples include antigen–antibody, lectin–sugar, streptavidin–biotin, and Protein A/G with the Fc region of the antibody. Since this natural immobilization occurs at high concentrations but with high sensitivity, this process is useful for microarray applications [35].

Irreversible Immobilization: The biosensor molecules irreversibly adhere to the sensor matrix.

Cross-linking: Here, biomolecules are linked to the transducers via cross-linking molecules with free reactive ends that form covalent bonds with functional groups. Whole cells and enzymes can be immobilized on the electrodes by being dipped in cross-linkable polymers such as glutaraldehyde and glyoxal.

Covalent binding: This binding is the most exploited, is stable but generally irreversible, and is stronger. It is achieved through functional groups like amino, hydroxyl, carboxylic, and thiol. It provides a homogeneous and dense distribution of the sensor molecules on the sensing surface to improve sensitivity. DNA immobilization with covalent bonding generally involves linker functional groups. Due to its specificity, biomolecular leakage is evaded here. Examples include 1-ethyl-3-(3-dimethyl aminopropyl) carbodiimide (EDC), a carboxyl activating agent, and N-hydroxysuccinimide (NHS) for coupling carboxyl groups to primary amine groups, yielding reactive NHS esters [12].

Entrapment: Here, the biosensing molecule is enveloped within a polymeric network while allowing substrates and products to pass through. This is a simple way to encapsulate, absorb, or localize the analyte molecules into the sensor matrix. It is achieved either by electrochemical polymerization of the polymer on the transducer surface or layer-by-layer deposition [12]. Entrapment is used for trapping cells or cellular organelles into the pores of the membranes. Enzymes trapped within reverse micelles remain effective during the fabrication of biosensors [36]. For example, sol–gel-like polymers, chitosan, starch, and polyacrylamide are used as encapsulating agents to provide an optimum environment for the molecule [37].

## 6. Biosensors for *Candida* Detection

Biosensors are classified on the basis of the signal transducers involved. Accordingly, these may be electrochemical, piezoelectric, optical, thermal, and surface plasmon resonance-based. Here, we describe the biosensor techniques devised for the detection of *Candida* species (Figure 2). The biosensor mechanisms for diagnosing *Candida* species are summarized in Table 1.

### 6.1. Electrochemical Biosensor

Electrochemical biosensors have gained popularity due to their rapidity, simplicity, and ample scope for miniaturization. Electrochemical biosensors may be amperometric, voltammetric, or potentiometric, based on the type of output.

*C. albicans* has been detected by an electrochemical personal glucose meter. Cell walls of *C. albicans* break down to form glucose when the cell-wall-degrading enzyme hemicellulose is added to the infected sample. The level of glucose is proportional to the cell number in the sample. A personal glucose meter has been repurposed for detecting and quantifying the amount of glucose. In these meters, glucose in the blood reacts with the enzyme electrode (glucose oxidase), leading to a sequence of oxidation and reoxidation reactions involving a ferrocene derivative and ferricyanide ions generating an electric current [39]. Similarly, an anti-*Candida* antibody-functionalized sensor specifically, accurately, and rapidly detects the presence of *Candida* using membrane-based electrochemical impedance spectroscopy. The sensitivity of the biosensor was up to the clinically relevant level of 10 CFU/mL [60].

Sensitive and fast detection of *Candida* species—*C. albicans*, *C. tropicalis*, *C. krusei*, and *C. glabrata*—based on their ploidy has been demonstrated by an electrochemical AMP-based biosensor. The sensor platform is composed of self-assembled nanofilms of electropolymerized poly(thiophene acetic acid) (PTAA) and amino-functionalized TiO_2_ nanoparticles (NH_2_-TiO_2_ NPs) and presented the highest electrochemical response when binding was analyzed using the electrochemical impedimetric technique [38].

### 6.2. Piezoelectric Biosensor

A piezoelectric biosensor records the oscillation change due to the binding event on a piezoelectric crystal. A piezoelectric immunosensor for *C. albicans* was designed in 1986 by the Japanese group of Muramatsu et al. [40]. The detection range was 1 × 10^6^–5 × 10^8^ cells cm^−3^. Anti-*Candida* antibodies were immobilized on the surface of anodically oxidized palladium-plated electrodes. A quartz piezoelectric crystal sensor was dipped into the *Candida* suspension to measure the increment in surface mass due to immunoadsorption of the microbial cells and the decrement in the resonance [40].

### 6.3. Optical Biosensor

Optical biosensors are the most intuitive biosensors of all the categories. Here, biomolecular concentration in a test sample is measured by measuring the change in optical signals like reflection, refraction, diffraction, absorption, fluorescence, and chemiluminescence. In optical biosensors, optical transducers that detect the changing intensity of absorbed or emitted light proportional to that of quantum change in the biorecognition element are exploited. These have label-free molecules, high specificity, economic production costs, and a small footprint.

Two-dimensional arrays of photonic crystals embedded in Con A hydrogels selectively identify *C. albicans* with the detection limit of 32 CFU/mL, by binding to mannan on its surface. Two-dimensional particle spacing in the crystal is reduced due to cross-linking, causing a blue shift of the diffracted light. This visual diffraction shift is either measured with a spectrometer or determined from the Debye diffraction ring diameter.

Diagnosis of oral candidiasis in smears and swabs is rapidly performed by fungal fluorescent staining as well as with periodic acid–Schiff stain and observation under a fluorescence microscope [60]. The pathogen is identified by blue fluorescence surrounding its annular or tubular structures [61].

A portable, rapid multi-target system for nucleic acid analysis devices has been fabricated for the simultaneous detection of four *Candida* species—*C. albicans*, *C. parapsilosis*, *C. glabrata*, and *C. tropicalis*—involved in vulvovaginal candidiasis. The designed system involves a sample-processing cassette with a nucleic acid analysis device. Nucleic acids are released within 15 min by a vibrating injector and grinding glass beads and analyzed within 30 min. Specific primer sets were introduced for each species. The technique used is the loop-mediated isothermal amplification (LAMP) method. The sensitivity is <2 CFU/reaction with a very low sample size of 1.41 µL [62].

A lateral flow strip (LFS) is a very popular, conventional, portable, and rapid optical biosensor. To diagnose the rise in the incidence of *C. parapsilosis* infections, a reliable DNA detection assay has been developed by combining LFS with recombinase polymerase amplification (RPA). The sensitive and specific detection utilizes the catalytic subunit 2 of the β-1,3-glucan synthase (*FKS2*) gene of *C. parapsilosis*, amplified by a primer–probe set with base mismatches (one base modified by the reverse primer and four bases modified by the probe). The rapid amplification and the appearance of a pink test line of the FITC and biotin-labeled FKS2 gene on the strip takes only 30 min at 37 °C. The efficiency of the method is 5.0 × 10^2^ copies/50 µL per reaction [42].

### 6.4. Surface Plasmon Resonance (SPR)-Based Biosensor

SPR is an optoelectronic phenomenon where a ray passes through a specific prism and is collimated to a photodiode array detector at a definite refractive index. When the frequency of the incident radiation becomes equal to the resonance frequency of the metal, electrons from the metal surface move after gaining energy from the photon. This creates an electric wave called a plasmon. The SPR phenomenon is driven by the refractive index (RI) of the metal, and any binding event brings about a change in the RI of the metal surface, which in turn changes the reflection angle of the light. This class of biosensors provides fast, label-free real-time monitoring of analyte concentrations, with minuscule sample sizes and reusable sensor chips. These sensitive analyses can be performed in a short time with minimal reagents. This mechanism is tuneable with various materials like quantum dots, metallic nanoparticles, graphene, graphene oxide, nanocages, nanorods, spherical gold nanoparticles (AuNPs), nanocages, and nanorods [63].

A capture antibody against *C. albicans* was covalently immobilized on mixed self-assembled monolayers by 11-mercaptoundecanoic acid and 3-mercaptopropanol. Using SPR, the detection limit of direct detection of 10^7^ cells/mL was increased to 10^6^ cells/mL using the sandwich antibody. The specificity for *C. albicans* was noteworthy among other oral pathogens, viz., *Escherichia coli*, *Staphylococcus aureus*, *Streptococcus mutans*, *β-streptococci*, and *Lactobacillus casei* [43]. A localized SPR surface-based molecular biosensing platform for targeting *C. albicans* was fabricated. Briefly, silver nanospheres functionalized with monoclonal anti-*Candida* IgG antibodies were tethered on a glass slide to detect pathogenic antigen concentrations as low as 50 ng/mL. At this concentration, a red shift of at least 8nm was observed in the spectrum, and it increased to 25 nm at 200 ng/mL [43].

An LSPR-based optical immunosensor platform for molecular biosensing has been engineered using anti-*Candida* monoclonal antibody-functionalized silver nanoparticles adhered on a glass slide. The binding of *C. albicans* antigens at very low concentrations of 50 ng/mL yielded an LSPR effect for nanosensing. Computational simulations, the spatial distribution of electromagnetic field enhancement around the nanoparticles, and molecular and bulk sensitivity were the parameters used for the analyses [44].

It is possible to upgrade SPR biosensors to miniature plasmonic nanobiosensors by modulating each chemical sample on the chip. Portability, real-time sample analyses, and the use of metamaterials are some of the issues of this category. Chemical stability, easy availability, lower costs, and thence commercialization may be achieved by replacing AuNps with TiNPs [64].

### 6.5. Nucleic Acid-Based Biosensors

Clinical diagnostics has taken a huge leap during the COVID-19 pandemic with reverse transcriptase PCR (RT-PCR) emerging as one of the primary tools due to its high sensitivity and low turnaround time. Other PCR-dependent tools are amplified fragment length polymorphism (AFLP), restriction fragment length polymorphism (RFLP), nested PCR, random amplification of polymorphic DNA (RAPD), multiplex PCR, and single-strand conformational polymorphism (SSCP) analysis [65].

The integration of PCR with other transducer modalities will hence lead to the development of point-of-care biosensors. *Candida* species can be identified by PCR amplification of their nucleic acid material with specific primers or pan-fungal primers from any infected sample. In the case of the latter, the products can be sequenced to achieve correct species identification [12,66]. An ultrasensitive biosensor for the detection of the fungal genomic DNA of *Histoplasma capsulatum*, causing serious pulmonary histoplasmosis, has been designed on the basis of three capture probes and three labeling oligonucleotide probes. Biotin-labeled probes were immobilized on streptavidin-coated 96-well plates [67]. A similar platform may be used to search for genomic DNA in infected samples. Quantitative PCR (qPCR) of the ITS2 region from spiked human blood and *C. albicans* culture showed a sensitivity of 10 CFU/mL and 0.2 CFU/mL, respectively [59,68]. The molecular identification of clinical *Candida* isolates has been reliably and accurately performed using real-time PCR followed by high-resolution melting analysis. This closed-tube powerful tool can detect changes even up to single-nucleotide polymorphisms, mutations, epigenetic information, and species identification by the comparison of the shapes and relative positions of the melting curves. In this way, eight species can be easily identified: *C. albicans*, *C. glabrata*, *C. parapsilosis*, *C. tropicalis*, *C. krusei*, *C. lusitaniae*, *C. kefyr*, and *C. guillermondii* [65]. Similarly, *C dubliniensis*, a predominant oral pathogen from cancer patients, could be identified along with *C. albicans* [69]. The polymerase spiral reaction of the ITS region between 5.8 S and 28 S of fungal ribosomal DNA could detect 6.9 pg/μL of *C. albicans* genomic DNA within 1 h [70]. A real-time PCR detection kit for *C. auris* has been developed by Biopremier.

The discussion of nucleic acid biosensors seems to be incomplete without the mention of clustered regularly interspaced short palindromic repeat (CRISPR)-based biosensors. Despite its shortcomings like a costly thermocycler and the need for trained professional technicians, CRISPR is one of the powerful nucleic acid tools for discovering new therapeutic targets and pinpointing, understanding, and possibly avoiding the emerging drug resistance spectrum in *Candida* species, e.g., azole resistance in *C. auris* and caspofungin resistance in *C. albicans*, *C. tropicalis*, and *C. parapsilosis* [71]. Dubey et al. have rightly remarked, “The efficacy and versatility of CRISPR/Cas9 have the potential to have a significant impact on the medical microbiology, both within the research facility enhancing genetic manipulation for investigating bacteria and fungi physiology and pathophysiology, deconstructing the function of virulence genes, and discovering different potential therapeutic targets and host–pathogen interaction—and at point-of-care enhancing the prognosis of fungal infected individuals” [72]. Different read-out methods like mobile phones, electrochemical-based methods, colorimetry, LFB, fluorescence, and paper-based assays may be used in conjunction with CRISPR diagnostics.

### 6.6. Nanomaterial-Based Biosensor

Nanomaterials are game-changers in the field of biosensors. They have tuneable physicochemical and electrochemical properties and hence can be functionalized in several ways. They have a high surface-to-volume ratio, good biocompatibility, chemical stability, and miniaturization potential. Their electronic performance can be modulated by doping and modulating energy gap distributions. Due to their very close energy levels, an infinitesimal change in an energy level is promptly detected, leading to high sensitivity towards bioanalytes. Therefore, they can be tracked by optical (UV-visible, fluorescence, X-ray photoelectron), thermal (temperature change), physical (shape change in the presence of stress), and electronic means. The shape modulation of nanomaterials into nanoparticles, nanosheets, nanowires, nanotubes, nanocages, and nanoarrays is another advantage.

Sá et al. have developed an innovative, simple, and sensitive lectin-based electrochemical impedance biosensor suitable for species identification of *Candia* species like *C. albicans*, C. krusei*, C. parapsilosis*, and *C. tropicalis*. Characteristic impedimetric responses were obtained due to the selectivity of lectins, either concanavalin A or wheat germ agglutinin-modified gold nanoparticles on the cysteine monolayers, towards carbohydrate moieties on the yeast cell wall. Gold nanoparticles have good oxidative stability and a very low Dru. The charge transfer resistance was proportional to *Candida* CFU. The detection limit was within 10^2^ to 10^6^ CFU/mL, which was evaluated by electrochemical impedance spectroscopy and atomic force spectroscopy. The sensitivity of the ConA-based biosensor was *C. parapsilosis* < *C. tropicalis* < *C. albicans* < C*. krusei*, and that of the agglutinin-based biosensor was *C. tropicalis* < *C. albicans* < *C. parapsilosis* < *C. krusei* [47].

An immunosensor for *C. albicans* detection has been designed based on a single-walled carbon nanotube (SWCNT)-mediated electron transfer process [73]. Neely et al. developed a portable, bedside nano-inspired platform for rapid tracking of candidemia pathogens—*C. albicans*, C. *parapsilosis*, *C. krusei*, *C. tropicalis*, and *C. glabrata—*in a patient’s blood. Nanoparticles tagged with complementary “capture probes” were allowed to bind with amplified pathogenic DNA obtained after lysis. Nanoparticles with complementary “capture probes” could then bind to the amplified DNA. Detection of up to 1–2 CFU/mL was performed with T2 magnetic resonance without any false positives [46].

Rhodamine B, a fluorescent indicator filled into a nanoporous anodic alumina scaffold with a grafted biomolecule, e.g., DNA/oligonucleotides, acts as a “molecular gate” for specifically recognizing *C. auris* genomic DNA at levels as low as 6 CFU/mL (Figure 3). This biosensor is selective, fast, and sensitive and does not need DNA extraction or amplification. The results are ready within one hour [74]. Very similar technology had earlier been explored for *C. albicans* [48].

T2*Candida* is the first FDA-approved, automated, nano-diagnostic panel for candidemia diagnostics from blood samples. The patient’s blood is collected in a K_2_ EDTA vacutainer collection tube. RBC lysis, debris collection, fungal cell collection, lysis, DNA amplification with specific primers, and the use of thermostable polymerase for ribosomal DNA intervening transcribed spacer region 2 are all performed in the automated instrument platform T2Dx. The amplified product is detected by T2 magnetic resonance of an amplicon-induced cluster of super-magnetic particles. It is quite effective for the detection of *C. albicans*, *C. glabrata*, *C. krusei*, *C. tropicalis*, and *C. parapsilosis* within the 1–3 CFU/mL range. The benefit of this method is the non-amplification of non-cell-associated, freely circulating DNA [75].

In a recent application, *C. albicans* was detected with poly-dopamine-co-chitosan composite gel-modified copper sheet microelectrodes integrated into a biochip. The composite gel allowed the adhesion of microbes leading to 99.9% enrichment while the electrical impedance spectrum quantitatively detected the concentrated microbes up to the single-cell level. Peak identification for each type was established by the principal component analysis method [50].

In an interesting development, a non-destructive organic–inorganic hybrid nanocatcher has been used for the ultrafast capture of invasive *Candida* specie*s*. A complex of magnetic nanoparticles -N-isopropylacrylamide-acrylic acid-caspofungin (MNP-NIPAM-AA-CAS) compact shell layers prepared via ultrasound-induced polymerization attaches specifically to the fungal cell walls present in the complex samples within 10 s. Subsequently, fungal cells are identified by surface-enhanced Raman spectroscopy (SERS) within 10 min. NIPAM is thermoresponsive, AA is the coupling unit, and magnetic nanoparticles and caspofungin offer structural stability and specificity, respectively. Its sensitivity is 10^2^ cells/mL. Identification is performed by determining differences in cell wall components of each strain, depicted as different peak values in the Raman spectrum [51].

A novel, easy, rapid, low-cost LAMP combined with nanoparticle-based lateral flow biosensor (LAMP-LFB) assay has been developed for *C. albicans.* The genomic DNA of *C. albicans* was amplified at 64 °C for 40 min by a set of six primers which were designed based on *C. albicans* species-specific internal transcribed spacer genes. The results from LAMP were reported visually by LFB within 2 min with a sensitivity as high as 1 fg of genomic DNA. Since there were no cross-reactivities with the non-*albicans* strains, the analytical specificity was completely 100%. The whole procedure was completed within 85 min (sample processing time, 40 min; amplification reaction time, 40 min; and result time, 2 min). The diagnostic accuracy was 100% [75]. Another multiple cross-displacement amplification of ITS-II assay, in combination with the AuNP-based LFB visualization method, has been used for *C. albicans* targeting. The sensitivity is 200 fg of genomic DNA [52].

Recently, an electrochemical, highly reusable, *C. auris* genosensor has been fabricated using ninhydrin as the DNA hybridization indicator by Guedes et al. Ninhydrin is an economical electrochemical DNA hybridization indicator with low oxidation potential and an environment-friendly molecule. It shows an increase in the oxidation peak during duplex formation. Twenty-five-base-long oligos were used as capture probes to locate *C. auris* genomic DNA in the urine samples. The detection level was 4.5 pg μL^−1^ [53].

A multiplexed plasmonic optical nanosensor array has been designed for the detection of pathogenic fungal DNA sequences. Arrays of microspots of capture DNA sequence-functionalized gold nanoparticles bind with the target sequence at the functionalized spot. The capture sequences are complementary to the pathogenic analyte DNA. The resultant change in the local RI due to the binding event is detected as a peak shift in the local SPR spectrum. Non-specific binding was minimized by passivation with herring sperm DNA and coadsorption with mercaptohexanol. The sensor can be regenerated by dehybridization with 10 mM HCl [54].

As a step towards the next generation of electronic biosensors, the polyclonal Systematic Evolution of Ligands by EXponential enrichment (SELEX) aptamer library has been utilized for the differential identification and fluorescence labeling of both non-albicans and albicans *Candida* pathogens like *C. parapsilosis*, *C. auris*, and *C. albicans* (Figure 4). The single-stranded aptamer library is stepwise incubated with the counter target (human cells) and target (*Candida*). Eluted bound aptamers are PCR amplified with phosphate-labeled primers and Cy5. After strand separation, followed by repeated (eight) rounds of target binding and amplification, these are subjected to analytical techniques of fluorescence microscopy, the fluorometric assay for fluorescent labeling, and flow cytometry [55].

The ability to colonize various parts of the human body is another dangerous feature of *Candida*. Its biofilm-forming potential is aided by quorum-sensing molecules like tryptophol, aromatic alcohols, farnesol, tyrosol, and phenylethanol. Combining 10% each of MWCNTs and crown ether, 12-crown-4-ether, improved the selectivity and sensitivity of tryptophol detection to 1 μg/mL. This combination yielded the highest oxidation response due to only one electrochemically active phenolic active site of tryptophol [56].

Recently, a nanobiosensor has been reported for *Aspergillus galactomannan* analyses [76]. Such studies may be extended further in the field of *Candia* diagnosis.

## 7. Emerging Biosensor Methods for *Candida*

Molecular diagnostics are novel and emerging biosensing methods that work on various microorganisms, including the detection of *Candida* sp. [77,78]. The FISH method has been successfully exploited, alone or in combination, for *Candida* surveillance. PNA FISH probes hybridize with target DNA with high specificity, sensitivity, and strength, primarily due to their structure’s electrical neutrality (absence of phosphate groups). Hybridization occurs rapidly (within a couple of hours) with low background noise. Faster species identification has been achieved with Yeast Traffic Light peptide nucleic acid FISH (PNA-FISH). Here, yellow light is generated for *C. tropicalis*, green light for *C. albicans* and *C. parapsilosis*, and red for *C. krusei* and *C. glabrata* [57]. In a clinical study, *Candida* species were rapidly identified from blood and peritoneal fluid cultures by PNA-FISH [79].

Silicon-based organic polymeric microfluidic hydrodynamic cell trapping has been integrated with PNA-FISH candiduria diagnostics. It offers direct target visualization amongst complex biological matrices with an epifluorescence microscope within a very short pre-enrichment time with a µL volume of samples. The FISH protocol was applied to trapped *C. tropicalis* cells which were permeabilized to internalize the hybridization probes [58].

Point-of-care (POC) devices are essential to reap the benefits of *Candida* biosensing. A POC device has five components—a target, a probe, a sensing module, a transducer, and a signal read-out device. A microfluidics-based detection system has been applied for *C. albicans* in blood [80]. Combining FISH with microfluidics has paved the way for the development of a future POC detection platform.

The internal transcribed spacer (ITS), a part of the ribosomal RNA gene complex, has been targeted for candidemia surveillance by quantitative PCR displaying a sensitivity of 0.2 CFU/µL. This region is present in 55 copies/haploid genome. It generates quantitative data to ease the evaluation of the magnitude of *C. albicans* infection in human blood [59].

Another method, MALDI-TOF, is now being gauged as one of the diagnostic procedures for patient samples. MALDI-TOF mass signatures are used to identify the clinical fungal species as well as the associated drug resistance. The MALDI-TOF spectrum peaks are compared with signature peaks from reference spectra that are contained in a database. It is one of the fast, accurate, and straightforward methods of identification [81]. In fact, *C. orthopsilosis*, *C. metapsilosis*, *C. parapsilosis*, *C. glabrata*, and *C. bracarensis* species could not be identified by the biochemical approach but could be identified by MALDI-TOF MS systems [82]. However, the pitfall of this option is the non-availability of a database for emerging strains, the lack of expert technical know-how, and the need for expensive and bulky machinery. The acquisition of correct fingerprint peaks depends on the sample preparation and operating settings of the instrument.

## 8. Future Perspectives and Directions

The development of biosensors for the detection of *Candida* species was initiated as early as 1986 [40]. This method required a minimum of 30 min of contact with the cells and was able to detect 10^6^–5 × 10^8^ cells cm^−3^. However, there was a gap in this direction until 2013, when gradually studies picked up. Further studies in this direction received a boost from 2021 onwards, perhaps coinciding with the emergence of COVID-19 detection systems (Figure 5).

Modern-day research aims for molecular diagnostic POC biosensors rather than conventional ones. For enhancing specificity, biomarkers for *Candida*, especially cell wall components, should be targeted for innovative biosensor development. Other factors in the diagnosis process should also not be overlooked, including the clinical state of the patient and other possible risk factors associated with higher invasive candidiasis rates, such as surgery, trauma, bacterial or viral infections, overuse of antibiotics, and catheterization [83,84,85].

Future studies should be directed towards multiplexing for high-throughput screening, refinement of sensor precision to the pico-/femtogram level, cost-effectiveness, portability, rapidity, and user-friendliness (Figure 6).

Enhancing the sensitivity and specificity of aptamer-based biosensors with nanomaterials creates intriguing new opportunities for a variety of applications, including the detection and analysis of fungal infections in clinical and environmental samples [86,87,88]. Smartphones and aptamers have been combined to create sophisticated, convenient, portable, and affordable in situ PON/POC biosensors for the detection of biomolecules. These sensors are ideal for a number of uses, including environmental monitoring and food safety, as well as disease diagnosis including *Candida* infections [89].

Advances in nanotechnology, CRISPR technology, and electrochemical and optical- or spectra-based biosensing methods have already been kickstarted; however, these technologies are yet to foray into *Candida* biosensing. Hence, a multi-disciplinary approach involving biology, chemistry, materials science, microelectronics, nanotechnology, computer science, and data science is necessary in this direction. The design of AI-assisted biosensors and precision medicine will boost the means to tackle the *Candida* infection menace. As in the case of bacterial infections, the application of machine learning models, AI tools, and neural networks to big biomedical cloud databases generated from infection with different *Candida* species, drug resistance and susceptibility, the emergence of resistant strains, and culture methods will further assist in the development of specific biosensors [90,91,92].

Genomics and biomarkers will impact final clinical judgments. The integration of the Internet of Things with biosensors will facilitate remote healthcare systems and personal medicine. This approach will lead to expeditious, continuous real-time monitoring of the circulating analytes reflecting the efficacy of antifungal treatment.

Some challenges in regard to the use of biosensors in microbial detection include the need for an in-depth assessment of how these machines perform in terms of false negative and false positive identification, environmental influences (temperature, pH, humidity, or electromagnetic fields), the need for sample preparation, and the need for regular and careful calibration [93].

## 9. Conclusions

The use of biosensors offers a smart solution for rapid and accurate detection, with various biosensor platforms and emerging technologies that can improve point-of-care diagnosis being available. The use of biosensors can revolutionize the diagnosis of *Candida* infections, as traditional diagnostic methods are time-consuming and expensive and require skilled personnel, while biosensors offer rapid, accurate, real-time detection of *Candida* species. The use of biosensors can also reduce the cost of diagnosis and improve patient outcomes. Electrochemical biosensors are the most commonly used biosensors for the detection of *Candida* species. They are simple, cost-effective, and offer rapid detection. Optical biosensors are also promising for the detection of *Candida* species. They offer high sensitivity and specificity and can detect multiple analytes simultaneously.

Nanobiosensors are emerging biosensors that offer high sensitivity and specificity, with the capability to detect *Candida* species at very low concentrations. Emerging technologies that can improve point-of-care diagnosis for *Candida* infections include microfluidics, lab-on-a-chip, and smartphone-based biosensors. Microfluidics and lab-on-a-chip technologies offer miniaturized biosensors that can be used for point-of-care diagnosis. They are portable and easy to use and offer rapid detection. Smartphone-based biosensors are also promising for point-of-care diagnosis and offer a low-cost solution for rapid and accurate detection of *Candida* species.

In conclusion, biosensors offer a smart solution for the diagnosis of *Candida* infections, being fast, accurate, and cost-effective. These technologies offer miniaturized biosensors that can be used for point-of-care diagnosis, proving portable and easy to use and offering rapid detection. The use of biosensors can revolutionize the diagnosis of *Candida* infections and improve patient outcomes. Further research is needed to optimize biosensor platforms and emerging technologies for the detection of *Candida* species.

## Figures and Tables

**Figure 1 life-13-02099-f001:**
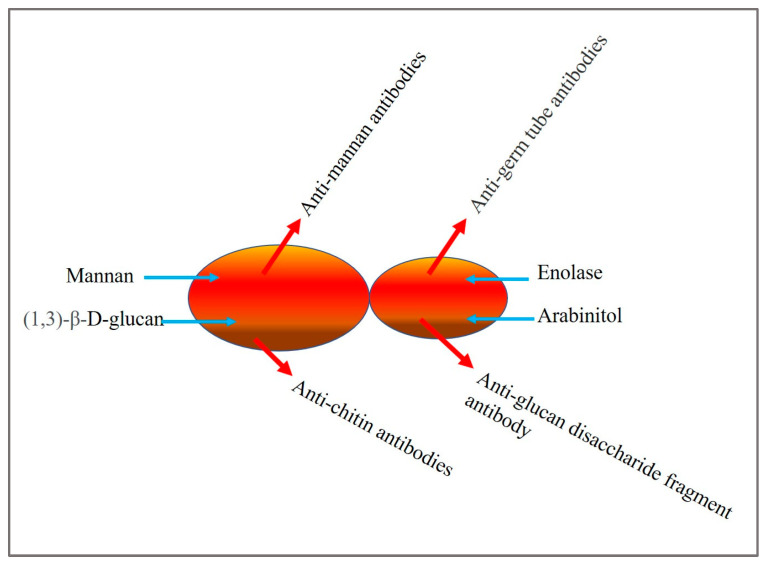
*Candida* biomarkers used for detection.

**Figure 2 life-13-02099-f002:**
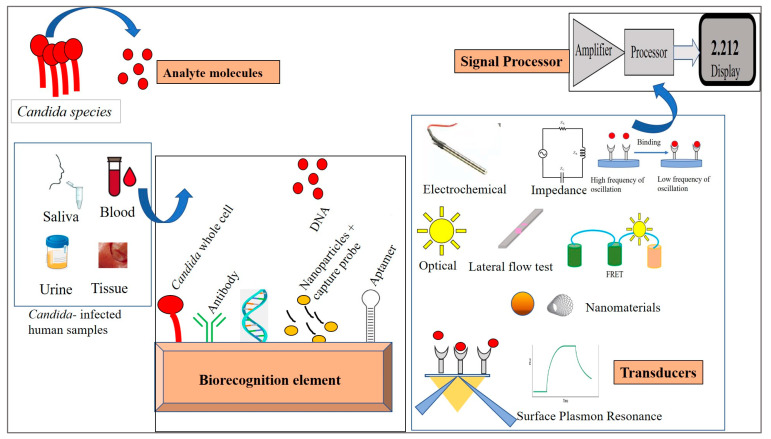
Different biosensors for the detection of *Candida.*

**Figure 3 life-13-02099-f003:**
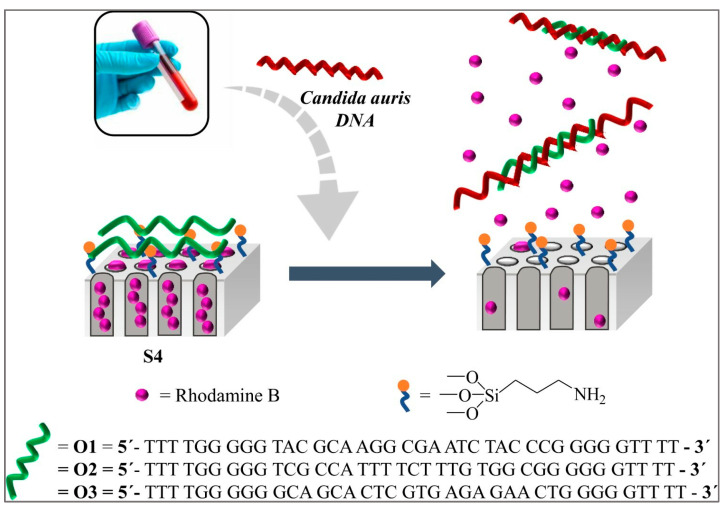
Schematic representation of the gated NAA support and the selective uncapping and dye delivery in the presence of *C. auris* genomic DNA (image courtesy of Pla et al., 2021, under Creative Commons CC BY license [74]).

**Figure 4 life-13-02099-f004:**
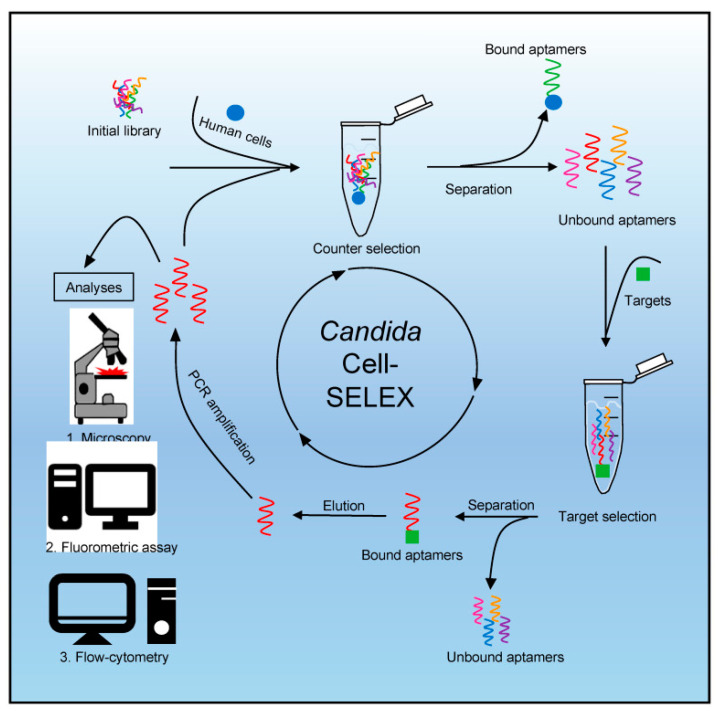
SELEX process and analytical techniques. The process starts with a counter selection where an initial single-stranded (commercial) aptamer library (TriLink BioTechnologies, Inc., San Diego, CA, USA) with approximately 6 × 10^14^ individual molecules is incubated with human cells as the counter target. Bound aptamers can be removed by a washing step to exclude them from further selection. The focused library is used in a target selection by incubating it with target cells (*Candida*). Aptamers that do not bind to target cells are removed by washing. The aptamers specifically bound to the cells can now be eluted, amplified by polymerase chain reaction (PCR) using Cy5 and phosphate-labeled primers, and then subjected as single-stranded molecules to different analysis techniques after strand separation, which is achieved by strand digestion with lambda exonuclease. This process is repeated to obtain a specific polyclonal aptamer library for *Candida* after sufficient rounds of target binding and PCR-mediated amplification. Analytical techniques include (1) fluorescence microscopy, (2) fluorometric assay, and (3) flow cytometry. Image courtesy of Kneißle et al., under Creative Commons Licence [55].

**Figure 5 life-13-02099-f005:**
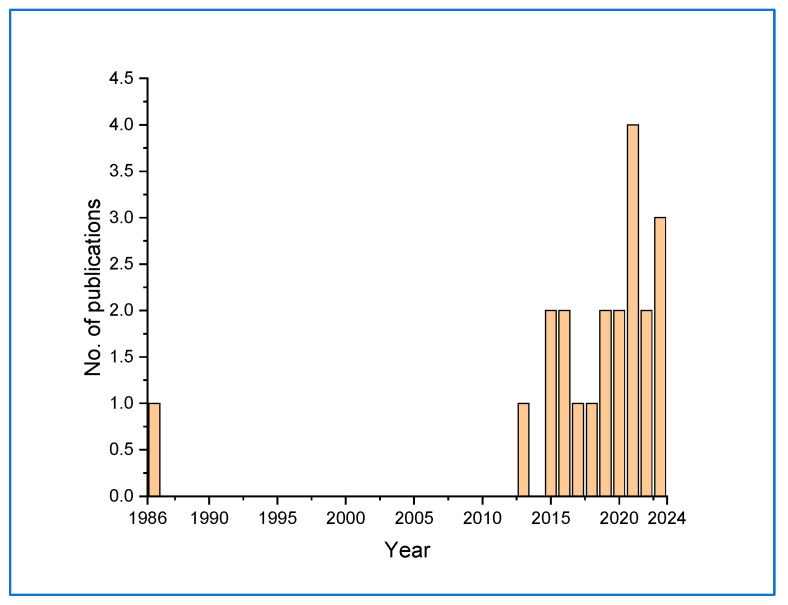
Graph showing the gradual development of *Candida* detection systems from 1986 to 2023.

**Figure 6 life-13-02099-f006:**
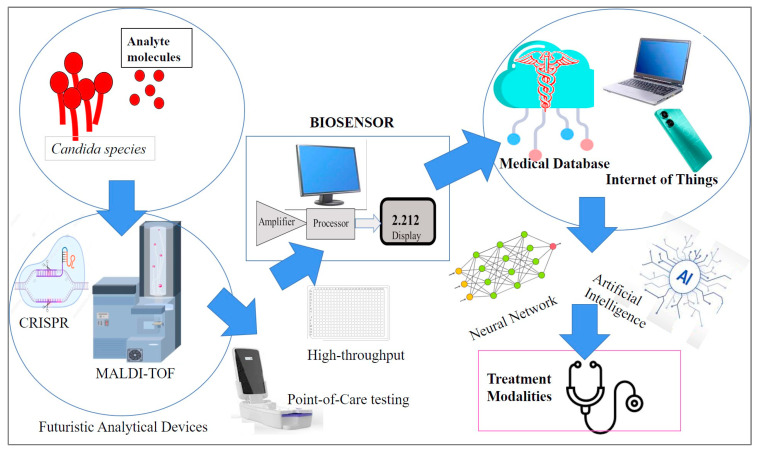
Futuristic biosensor concepts to be explored for *Candida* detection.

**Table 1 life-13-02099-t001:** Summary of biosensors applied for the detection of *Candida* species.

Biosensor Type	Sensor Material	*Candida* Species	Sample	Sensitivity	Reference
Electrochemical impedance	Electropolymerized poly(thiophene acetic acid) (PTAA) and amino-functionalized TiO_2_ nanoparticles	*C. albicans*, *C. tropicalis*, *C. krusei*, and *C. glabrata*	Culture	2 and 3 CFU mL^−^¹	[38]
Electrochemical impedance	Personal glucose meter	* C. albicans *	Culture, urine, serum, blood	10 CFU/ml	[39]
Piezoelectric immunosensor	Piezoelectric crystal	* C. albicans *	Culture	10^6^–5 × 10^8^ cells cm^−3^	[40]
Optical	2D arrays of photonic crystals	* C. albicans *	Culture	32 CFU/mL	[41]
Loop-mediated isothermal amplification (LAMP) method	Nucleic acid	*C. albicans*, *C. parapsilosis*, *C. glabrata*, and *C. tropicalis*	Vaginal Swabs	<2 CFU/reaction	[6]
Lateral flow strip	DNA	*C. parapsilosis*	Clinical samples	5.0 × 10^2^ copies/50 µL	[42]
Surface plasmon resonance (SPR)	Antibody	* C. albicans *	Culture	10^6^ cells/mL	[43]
SPR	Monoclonal antibody-conjugated AgNPs	* C. albicans *	*C. albicans* antigens	50 ng/mL	[44]
Impedance	Lectin-modified AuNPs	*C. albicans*, *C. krusei*, *C. parapsilosis*, and *C. tropicalis*	Culture	10^2^ to 10^6^ CFU/mL	[45]
Nanoparticle-mediated immunosensor	Single-walled carbon nanotubes (SWCNTs)	* C. albicans * , C. *parapsilosis*, *C. krusei*, *C. tropicalis*, and *C. glabrata*	Whole blood	1–2 CFU/mL without any false positives	[46]
Nanoporous anodic alumina nanogate	DNA/ oligonucleotides	* C. auris *	Blood	6 CFU/mL	[47]
T2 magnetic resonance assay	Magnetic resonance	*C. albicans*, *C. glabrata*, *C. krusei*, *C. tropicalis*, and *C. parapsilosis*	Whole blood	<1 CFU/mL	[48]
Electrical impedance	Poly-dopamine-co-chitosan composite gel-modified copper sheet microelectrodes	* C. albicans *	Culture	99.9%	[49]
Surface-enhanced Raman spectroscopy	Magnetic nanoparticles -N-isopropylacrylamide-acrylic acid-caspofungin	* Candida * species	Clinical samples	10^2^ cells/mL	[50]
LAMP-LFB	Nanoparticles and DNA	* C. albicans *	Isolated from clinical samples	1 fg	[51]
LFB	AuNPs	* C. albicans *	Isolated from clinical samples	200 fg	[52]
Genosensor	Ninhydrin-DNA	* C. auris *	Human urine enriched with *C. auris* gDNA	4.5 pg μL^−1^	[53]
Plasmonic optical nanosensor	Arrays of DNA sequence-functionalized gold nanoparticles	* Candida *	DNA target sequence	60 nM	[54]
Fluorescence and flow cytometry	SELEX Aptamer library	*C. parapsilosis*, *C. auris*, and *C. albicans*	Culture	0.2 ng of DNA and ~75% of the *Candida* cells	[55]
Electrochemical	MWCNTs and crown ether, 12-crown-4-ether	*Candida species*	Culture	1 μg/mL tryptophol detection	[56]
Yeast Traffic Light PNA-FISH	Fluorescence in situ hybridization	*C. tropicalis*, *C. albicans*, *C. parapsilosis*, *C. krusei*, *C. glabrata*	Blood	82.6% positive results	[57]
Microfluidic hydrodynamic cell trapping and epifluorescence	PNA_FISH	* C. tropicalis *	Culture, artificially contaminated urine sample	Error 5.38–10.75%	[58]
Q-PCR	Ribosomal RNA gene complex	* C. albicans *	Blood	0.2 CFU/µL	[59]
Electrochemical impedance spectroscopy sensor	Polycarbonate membrane	*C. albicans*	Culture		[60]

## Data Availability

Data are available upon request from the corresponding author.

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
