# Peer review of "Innovative Biosensing Approaches for Swift Identification of Candida Species, Intrusive Pathogenic Organisms"

_life, 2023, doi:10.3390/life13102099_

Round 1
Reviewer 1 Report
The authors gave an insight into different biosensor technologies designed for the detection of medically significant Candida species, especially Candida albicans and C. auris, and their applications in the medical setting. The work is interesting but should be better presented. I encourage authors to resubmit the manuscript after these corrections have been made.
Extra details about therapy are not needed in the introduction. What makes your review different from the already published reviews in this area or why this review paper is needed?
Add a few lines on the disadvantages of conventional approaches used for the identification of candida species, if they have any? What are the advantages or why preferring these biosensors over traditional methods to detect candida species.
Line 114 available literature time span should be mentioned.
There should be a timeline, graph, or an illustration to show how the development (in table 1) occurred gradually and if there is a limited data availability or gaps during the timespan explored by authors.
Carefully check the line 219.
Mention the biological fluid sample in the table for detection analysis.
Copy pastes marks are there in table in table 1, against reference 47, 48, 18, 31, and 51, 53, 57, 58.
Sections 7-12 should represent the subsections of section 6 instead of appearing as individual sections.
Mention the reason for your claim in line 587.
Some latest good quality publications in the area of biosensors, nano-biosensors and aptamer-based biosensors can be cited e.g., Environmental research (2023): 117123; Progress in Materials Science 129 (2022): 100967; Biosensors and Bioelectronics 215 (2022): 114509; Frontiers in Oncology 11 (2021): 632165; Nanomaterials, 11(4), 840.
There are some serious typos e.g., SHELEX (table 1, page 8),
Order of information need corrections in introduction. The overall flow of information is disrupted in the first few sections of the manuscript. Please re-arrange some sentences where they make a good flow to increase the readers interest. Remove the duplicated information to make this review more comprehensive and interesting.

Moderate editing of English language required
Author Response
Thank you for your comments, suggestions and support.
Please find the responses in the attached file.

Reviewer 2 Report
This review aims to give an insight into medically important Candida surveillance by different types of biosensors based on the available literature, emphasizing the impact of modern technologies, and their applications in the medical setting.It would be better if a specific biosensing approache could be added.
Moderate editing of English language required
Author Response
Thank you for your comments and suggestions.
Please find the response attached.

Reviewer 3 Report
The authors did a fine job of summarizing the current biosensing technologies used to determine Candida species. The organization of the information is logical and the flow is smooth and easy to follow. It would be an informative and educational reference for readers who are new to or interested in the field.
Below are my comments for consideration.
1. All the biosensors discussed would need blood samples. Has any non-serological device/sensor been reported in the literature?
2. In the introduction, include a brief genealogy of biosensors for Candidas determination so that readers could understand the past, present and existing challenges of the field.
3. The major drawback of biosensors is false positive and negative identification. The authors ought to discuss this issue in the review. If data are available, the information should be included in Table 1.
4. Analysis time is one of the critical performance criteria. The information should be included in Table 1.
5. Table 1. Revise the sensitivity spec of Electrical Impedance. What does "99.9%" mean?
6. Table 1. Biosensor Type column. Some text associated with the second/third sensors should be removed.
7. Lines 600-615. Nothing wrong to share some perspectives, but the authors need to be more specific regarding how AI and ML could take bio-sensing technologies related to the detection of Candida species to the next level. The text here is too general to be meaningful.
8. Sections 6-12. For individual sensors, it is important to discuss the pros and cons so that readers could better understand and differentiate the development applications of those sensors.
9. Table 1. Include a column to summarize advantages and limitations of the sensors listed.
Minor editing of English language required.
Author Response
Thank you for your comments and suggestions.
Please find the responses attached.

Round 2
Reviewer 1 Report
The table 1 still shows many new copy paste marks needs to be removed. Accept the paper for publication.
Reviewer 3 Report
The authors have addressed my comments and revised the manuscript accordingly. Therefore, I would like to recommend the current version for publication.
Minor editing of English language required